# The Rostral Epidural Rete Mirabile: Functional Significance in Blood Flow Regulatory Mechanisms in Giraffe (*Giraffa camelopardalis*)

Marna S. van der Walt [1], Willem Daffue [2], Jacqueline Goedhals [3], Sean van der Merwe [4] and Francois Deacon [1,*]

1 Department of Animal-, Wildlife- and Grassland Sciences, Bloemfontein 9301, South Africa
2 Kroonstad Dierehospitaal, Kroonstad 9500, South Africa
3 Department of Anatomical Pathology and National Health Laboratory Services, Bloemfontein 9301, South Africa
4 Department of Mathematical Statistics and Actuarial Science, Bloemfontein 9301, South Africa
* Correspondence: deaconf@ufs.ac.za

**Abstract:** The distinctive long neck of the giraffe (*Giraffa camelopardalis*) entails functional difficulties brought about by the extended distance between the heart and the head. Blood must be circulated over 2 m from the heart to the brain against gravitational force. The natural movement of the head to ground level would result in a large volume of blood moving toward the brain with the force of gravity. Large blood volumes also rush to the brain during bulls' fighting (necking), rendering the giraffe susceptible to possible brain damage. The natural movement of the head from ground level to fully erect would result in blood moving away from the brain with gravitational force. The lack of blood perfusing the brain can cause fainting. The giraffe, however, suffers neither brain damage nor fainting. What adaptations do giraffes have to counteract these challenges? The aim of this study was to investigate the functionality of the rostral epidural rete mirabile situated just beneath the brain and its possible contribution to successful circulation in long-necked giraffes. The unique rostral epidural rete mirabile structure significantly contributes to counteract physiological challenges. Turns and bends characterize this structural arterial meshwork and subsequently an increased artery length through which blood flow must proceed before entrance into the brain, exerting resistance to blood racing to the brain when the head is lowered to the ground. The brain is supplied mainly by the maxillary artery through the carotid rete, with a rudimentary basilar artery not contributing to the brain's blood supply. The resistance to blood flow due to the structure and position of the rostral epidural rete mirabile when the head is in the upright position is counteracted by the unique carotid-vertebral anastomosis allowing immediate cerebral blood supply. The rostral epidural rete mirabile structure in giraffes is an essential feature balancing physiological difficulties arising due to the extensive heart-to-head distance and might fulfill the same function in other long-necked artiodactyls.

**Keywords:** giraffe; cerebral blood supply; rostral epidural rete mirabile

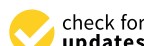

## 1. Introduction

Pragmatically, we can assume giraffes (*Giraffa camelopardalis*) experience several physiological challenges due to their extraordinary build. Detailed tests on captive giraffes showed that these animals have a very high blood pressure of approximately 200–400 mmHg [1–3]. At this pressure, certain key physiological issues arise. For instance, the heart must pump blood to the brain against gravitational pressure and would encounter vascular friction towards the brain, situated approximately 2 m above the heart. Some studies have investigated possible adaptations to the physiological challenges of giraffes [2,4,5]. Few studies describe the unified function of how giraffes successfully circulate blood, avoid fainting

and blackouts, escape brain damage and prevent oedema in the legs. A unique anatomical structural differentiation called the rostral epidural rete mirabile [4,6–9] is present in Cetartiodactyla, including giraffe. The rostral epidural rete mirabile is absent in Perissodactyls, but cats and dogs also have a rostral epidural rete mirabile [10–15]. The rostral epidural rete mirabile is described as a complex network of arteries and veins situated at the base of the cranium [8,9,16,17]. The entire rete in Artiodactyls is situated intracranially, within the cavernous venous sinus, which is different from other species, such as cats and certain primates that also have a rostral epidural rete mirabile [8,15,16,18]. Alteration of the intracranial segment of the internal carotid artery developing into the rostral epidural rete mirabile network structure occurs in the embryo. In Artiodactyls, the common carotid artery transitions into the maxillary artery, from where it enters the skull through the carotid foramen and transitions into the network of the rostral epidural rete mirabile. The brain blood supply is thus different from the supply via the internal carotid and basilar artery, as in other species [6,8], to the maxillary artery connecting to the carotid arterial network in giraffes. The rostral cerebral artery and the caudal communicating artery are responsible for the main supply to the cerebral arterial circle. In giraffes, the basilar artery is undeveloped and therefore does not form part of the pathways through which blood can be supplied to the brain [7,8,19]. The basilar artery in sperm whales (*Physeter macrocephalus*) is similarly non-functional concerning the blood supply to the brain [12]. The basilar artery is prominent in dromedary camels (*Camelus dromedarius*) and other ruminants and services the medulla oblongata, pons, and cerebellum [20]. The maxillary artery is dominant in its blood supply to the circle of Willis. Still, the basilar artery in these species is directly connected to the circle of Willis, with the course of flow directed caudally and not rostral [21]. Extending from the retial meshwork, the intracranial maxillary artery branch configures into the brain's arterial circle, also known as the circle of Willis [6,17,19]. The intracranial maxillary artery segment emerging from the rete forms a split with the caudal intracranial carotid artery [15]. The arterial circle of the brain receives blood predominantly from the maxillary artery with a non-functional basilar artery that cannot be utilized to supply any substantial amount of blood to the cranium in giraffes [8,15,19,21,22]. The study of [7] on Old and New World camelids showed that the cerebral arterial blood supply originates directly from the rostral epidural rete mirabile and the basilar artery. Additionally, in contrast to an artiodactyl's, the rostral epidural rete mirabile of camelids anastomose with branches of both the maxillary and internal carotid artery. [18] have suggested three functions of the rostral epidural rete mirabile: the regulation of blood flow and pressure to encephalic circulation, temperature regulation and the movement of pheromones from venous blood originating from the nasal mucosa to the hypophysis. [23] focused on the rostral epidural rete mirabile functioning to regulate blood flow in the average bodily environment. [10,14,24] described the rete to allow adequate brain perfusion in instances of break-in blood flowing towards the brain as a direct result of the various anastomoses between the two sides of the rete. The rete thus acts as a protective measure preventing high perfusion pressure with the facilitation of increased blood flow during unusual conditions. Furthermore, the rostral epidural rete mirabile allows for adequate blood flow necessary for brain function concurrently with reducing high systolic blood pressure, influencing mean perfusion pressure minimally [24], and the lack of valves within the cranial cavity support venous blood resuming from the brain. The morphology and structural function of the rostral epidural rete mirabile may vary in different species, with a distinction in the main blood suppliers to this meshwork [25].

This study points out the structure of the rostral epidural rete mirabile and its function in the long-necked giraffe. Furthermore, this study is part of a more significant study that aims to describe the combined features that allow giraffes to successfully circulate blood to the brain, avoid fainting, escape brain damage and prevent oedema in the legs.

## 2. Materials and Methods

Ethics approval nr. UFS-AED2020/0083. Approval was obtained from the Animal Ethics Research Committee at the UFS, SPCA and DESTEA.

### 2.1. Experimental Model and Subject Details

Samples were attained from animals culled by several nature reserves in South Africa and thus on an ad hoc basis. Unfortunately, the giraffe is culled by these reserves due to overstocking concentrations and diet shortages. The study samples were opportunistically gathered when these culling operations were conducted. The researchers were not involved in organizing or carrying out the culling procedure. As part of sustainable habitat use, most game ranches manage the number of animals conferring to the farm's carrying capacity (available diet) that gets evaluated yearly. This annual management procedure includes live trade, translocations or culling the excess animals. This process is a well-thought-out recurrent management method used in the Southern Africa Wildlife Industry to maintain herbivore numbers applicable to the carrying capacity of the farm or reserve and to evade death due to starvation [26].

### 2.2. Animal Sources

Five giraffes (*G. camelopardalis*), female n = 3 and male n = 2, were included in this study. Giraffe males weigh around 1200 kg with a total height of approximately 5.5 m. The female giraffe weighs around 830 kg with a total height of about 4.5 m [27]. Sample animals were obtained opportunistically on an ad hoc basis.

### 2.3. Method Details

A giraffe head severed between C3 and C4 cervical vertebrae was obtained. The vascular system was physically washed out with warm water. The specimen samples were raised with the nose pointing down and washed until the water was clear.

### 2.4. Dissection

Four giraffe heads were used for latex infusion. The latex mixture combinations used for the arterial and venous systems varied. The basic ingredients included: Barium sulphate ($BaSO_4$) (X-ray grade, Kyron Powder, Kyron, SA, and Latex moulding rubber (A. Shak (Pty) Ltd., Durban, South Africa). Additionally, to distinguish and differentiate the arteries from the veins, a red (Stamp ink, Office Mate, South Africa) pigment was used to infuse arteries, and a blue pigment (Print ink, Treeline, SA) was used in combination to infuse the veins. Arteries were infused with $BaSO_4$ powder at a forty percent volume, sixty percent latex volume and two percent volume red ink. Veins were infused with $BaSO_4$ powder of a twenty percent volume, eighty percent latex volume and two percent volume of blue ink. The vascular system was thoroughly rinsed with lukewarm water and subsequently filled with the different latex mixtures. A tube (10-mm PVW piping) was positioned 2 m above the vascular system to be able to use gravitation force to aid adequate latex infusion. Sutures (Catgut 3 Kyron, South Africa) and strings were utilized to keep the tubing in place. The common carotid artery was infused with the red combination latex, and the jugular vein with the blue combination latex. The infused arteries and veins were closed with catgut (Catgut 3 Kyron, SA). To encourage latex to set, the specimens were positioned at a 30° angle for 24 h. After the latex had been set, the specimens were separated at the midpoint between the ossicones with a Recipro saw (Makita). Care was taken to divide the specimens into identical halves through the skull and neck vertebra. Dissection originated at the C3 cervical vertebra trailing the arteries and veins up to the rete mirabile and the brain. $BaSO_4$ was included in the latex mixture for future radiographic imaging, as part of a dissertation.

### 2.5. Brain Endocast

Two giraffe heads were used to determine brain volume. Most of the skin and meat were removed from the skulls. The skulls were then buried and left to decay naturally, allowing a completely clean skull. The two halves of each head were filled with latex molding rubber (A. Shak (Pty) Ltd., SA) and allowed to set. Each latex cast was removed and paired with the correct other halves. Brain volume was determined by submerging the endocast in a measurement jug filled with water and measuring the amount of water displaced.

### 2.6. Histology Sample

One giraffe head was used for dissection to obtain histological samples of the carotid rete, the artery at the origin of the carotid rete and the artery at the exit of the carotid rete. The head was divided into two symmetrical halves, cut along the midline from the middle point between the two ossicones through the skull and neck vertebra with a Makita Recipro saw. The rostral epidural rete mirabile was located above the brain plate in the cranium and then photographed with a Canon 7D camera and a 24–105 mm lens. A sample of the rostral epidural rete mirabile was taken, as well as a sample at the origin before branching into the rostral epidural rete mirabile network. A sample was also taken at the exit of the rostral epidural rete mirabile network.

The samples were then placed in a 10% buffered formalin (10% aqueous solution and 4% formaldehyde). The formalin-fixed tissue was processed overnight in a VIP6 tissue processor and was embedded manually in wax blocks. Following embedding, 2–4 μm thick sections were cut, placed on glass slides, and stained with haemotoxylin and eosin (H&E) and Masson's Trichrome (for muscle and collagen fibers) and Verhoeff–Van Gieson (for elastic fibers) using standard methods. The stained slides were then covered, slipped and evaluated. Pictures were taken from the section pre-carotid rete, post-carotid rete and sections within the rostral epidural rete mirabile at a $2.5\times$ magnification with a Leica DM750 microscope with a Leica ICC50 W camera. The Tunica media of the arteries were measured using the microscope.

### 2.7. Pressure and Flow Tempo Experiment

We mimicked a basic rete network to determine what influence the number of turns or spirals in the rostral epidural rete mirabile structure might have on flow tempo. The serial setup of the experimental flow test, even though a limitation to the actual anatomy of the carotid rete, is utilized to represent the influence of the rete structure. A rostral epidural rete mirabile network was mimicked using 4-mm diameter plastic fish tank air pump tubing. The tubing was measured into 0.5 m, 1 m, 2 m, 3 m, 4 m, 5 m and 6-m lengths and then winded around a standard 25-mm diameter plastic pipe to imitate the rostral epidural rete mirabile (Figure 1a,b). Plastic fish piping was fixed against a wall, giving a length of 4750 mm. Water, at room temperature, was used. Connections were constructed in a manner so that water could be opened with a switch and for the water to be pumped through the rostral epidural rete mirabile and upwards to the height of 4750 mm. The water flow was opened for one minute. The amount of water that ran through the system was measured. The change in water level before and after the rete mirabile was measured after the minute (Figure 2A). The experiment was repeated five times for each of the different lengths that simulated the carotid rete.

Flow per minute was calculated by averaging the five measurements for each rostral epidural rete mirabile length. The pressure was calculated by using the same experimental setup. A pressure meter (Yoto PG802C-3-10KPA,2017101312,0-10KPA,0.25%FS) was inserted after the rostral epidural rete mirabile to measure pressure. The pressure difference was calculated using the pressure measured before the flow entered the rostral epidural rete mirabile network and after the flow through the rete mirabile network. The pressure measured post flow through the rostral epidural rete mirabile was subtracted from the pressure measured prior to entering the rete. The pressure before 4750 mmH$_2$O measured height gives the water column in meters, from where the pressure was calculated and

expressed in mmHg. The five pressure measurements were again averaged for each rostral epidural rete mirabile length (Figure 2B).

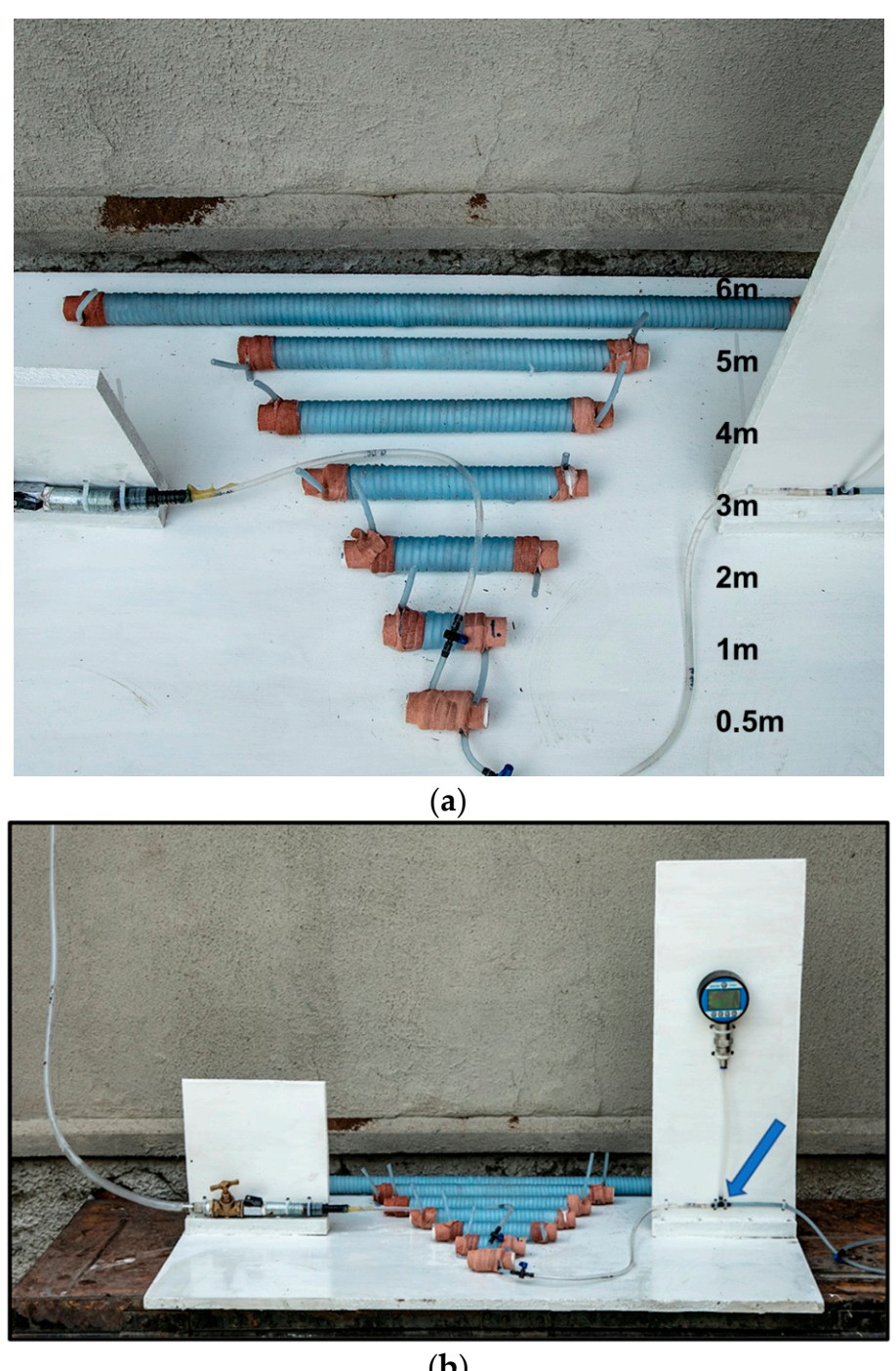

(**a**)

(**b**)

**Figure 1.** (**a**). Mimicked rostral epidural rete mirabile structures. (**b**). Mimicked rostral epidural rete mirabile network with pressure meter attached. Pressure is measured at the point indicated by the blue arrow.

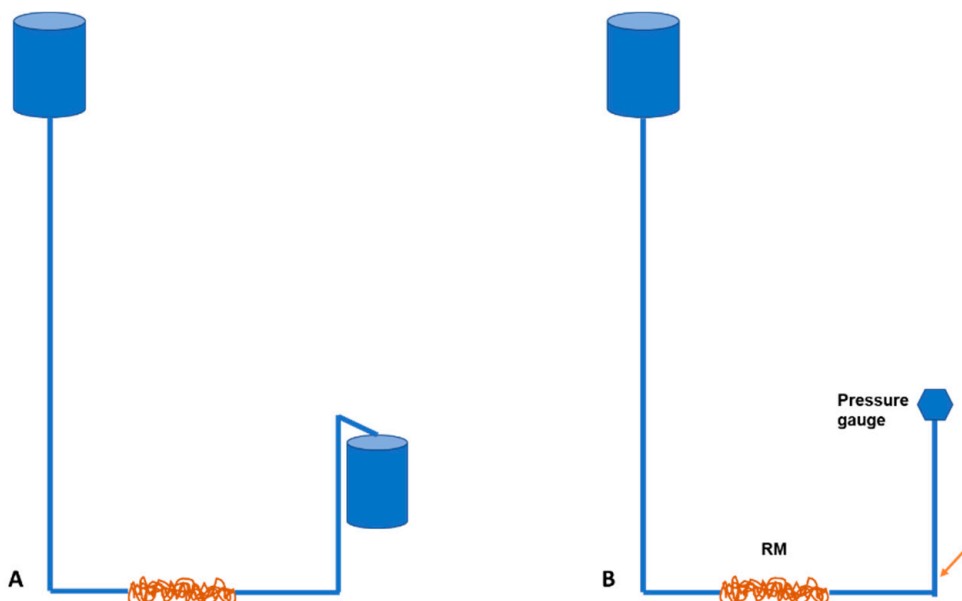

**Figure 2.** Schematic illustration of water flow (**A**) and pressure (**B**) experiment. Pressure is measured at the point indicated by the orange arrow.

## 3. Results

### 3.1. Dissection

During the dissection procedure, we first identified the exact location of the rostral epidural rete mirabile within the giraffe's skull. The location of the rostral epidural rete mirabile is in the cavernous venous sinus, above the brain plate within the cranium (Figure 3a,b).

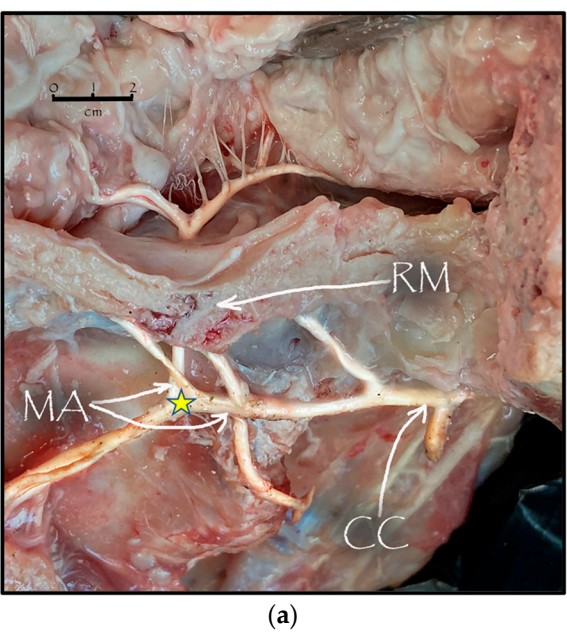

(**a**)

**Figure 3.** *Cont*.

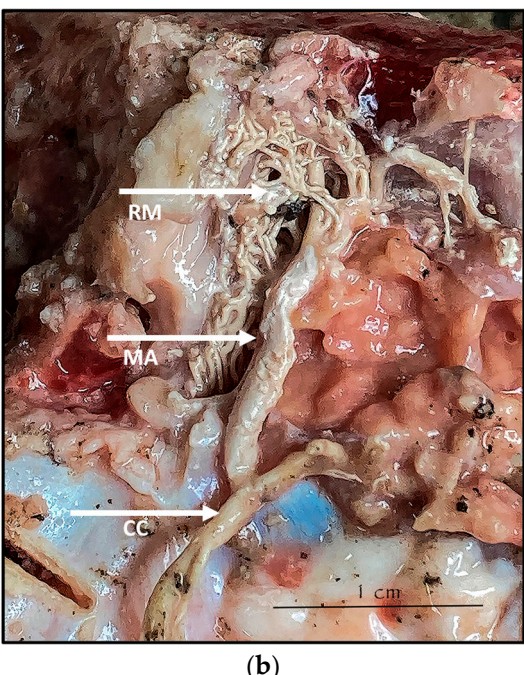

(**b**)

**Figure 3.** (**a**). The rostral epidural rete mirabile network is located in the cavernous venous sinus, above the brain plate within the cranium illustrating the rostral epidural rete mirabile (RM), the maxillary artery (MA) and the common carotid (CC). Notable bend towards the RM indicated by a yellow star. (**b**). Image illustrating rostral epidural rete mirabile (RM) structure below the giraffe brain, the maxillary artery (MA) and the common carotid (CC).

Upon dissection of the latex-filled arteries, the arterial pathway was followed from the C3 cervical vertebra to where the maxillary arteries enter the brain via the carotid foramina. The internal carotid artery forms the rostral epidural rete mirabile, the external carotid artery transitions into the maxillary artery. The pathway of the common carotid artery, with its transition to the external carotid artery and transition to the maxillary artery, is observed as a relatively straight pathway without bends. An interesting sharp bend is present before the entrance of the maxillary artery into the rostral epidural rete mirabile network (Figure 3a). Even though the latex injection did not perfuse much of the rostral epidural rete mirabile network, the network could clearly be identified, irrespective of size limitation, as visible in Figure 3b. Having access to the complete giraffe, we observed that the heart is situated anteriorly in the chest compared to the heart of other ruminants, such as elephants and sable antelope (Figure 4). The pathway of the blood from the heart to the brain is straight to allow for the easy and quick transport of blood without the resistance exerted by a curve or bend in the pathway.

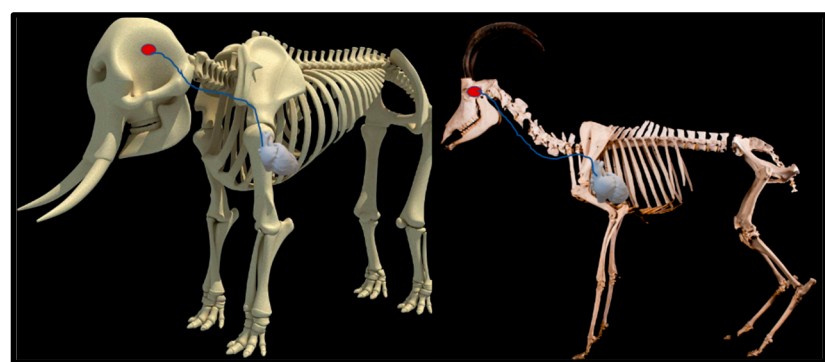

**Figure 4.** *Cont.*

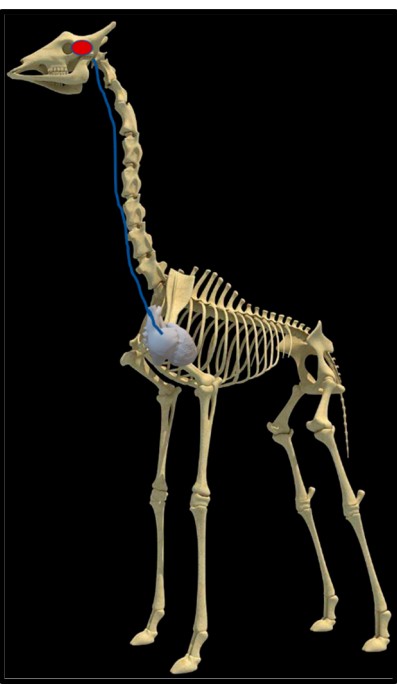

**Figure 4.** Position of the heart in a sable antelope (*Hippotragus niger*), elephant (*Loxodonta africana*) and giraffe (*G. camelopardalis*), with the blood flow pathway illustrated by the blue arrows, emphasizing the giraffe heart situated anteriorly in the chest.

We also confirmed that the basilar artery in the giraffe, unlike in other mammals, does not offer a second supply of blood to the brain [8,15,19,22]. The maxillary artery supplies the circle of Willis via the rostral epidural rete mirabile in the giraffe.

### 3.2. Brain Volume Endocast

The brain endocasts revealed that the male giraffe's volume was 900 mL and the female giraffe's 700 mL. In most mammals, the total volume of blood is 5.5–8% of body mass [28]. For our calculations, we will use an average of 6.75%. In humans, the total blood volume in the body is 5.7 L of body mass in males and 4.3 L in females on average [29]. The human brain weighs about 1500 g with 100–130 mL of blood present at any moment in the brain [30]. We used the average of 115 mL of blood at a specific time in a 1500-g brain for our calculations. Blood flow through the brain is thus 60 mL of blood/100 g/min, with a total of 900 mL of blood per minute in the brain of humans. In humans, blood needed in the brain equates to 2.02% of the total blood volume.

In this study, we found that for giraffes, the males have 69 mL of blood, and the females have 43 mL of blood present at any moment in the brain. This is notably a small volume compared to humans and other mammals. However, blood flow is relatively high, with 540 mL of blood per minute in the brain of a male giraffe and 420 mL of blood per minute in the brain of a female giraffe. If the brain's blood percentage is calculated in male (0.089%) and female (0.077%) giraffes, it equates to less than 0.01% of the total blood volume. Giraffes thus need an average of 24.46 times less blood in the brain, as a proportion of the total supply, compared to humans. Therefore, only a small amount of blood, 0.01%, must be present in the giraffe's brain at a specific time. Consequently, the physiological stress of large amounts of blood that need to be moved over the long head-to-heart distance to supply to the brain in the giraffe is reduced.

### 3.3. Histology

In Figure 5, an illustration of the section of the pre-carotid rete, within the carotid rete, and post-rostral epidural rete mirabile is shown. Pictures were taken from the Verhoeff–Van Gieson-stained sections at a 2.5× magnification with a Leica DM750 microscope with a

Leica ICC50 W camera. Measurements with the Leica DM750 microscope of the Tunica media of each of the sections are shown in Table 1.

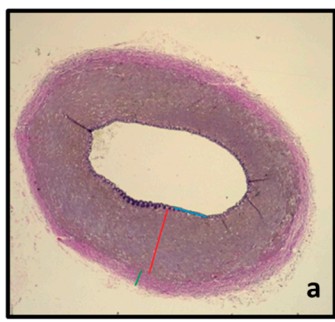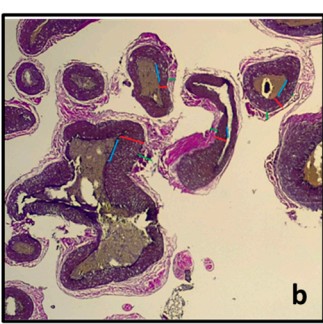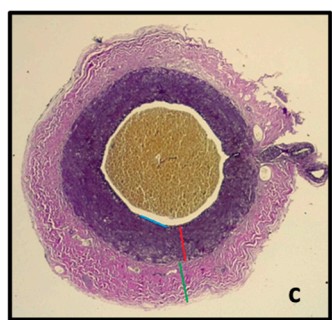

**Figure 5.** Section of the maxillary artery before entrance into the rostral epidural rete mirabile (**a**) showing the rostral epidural rete mirabile arteries (**b**) and the intracranial maxillary artery (**c**) with the T. intima (blue line), T. media (red line) and T. adventitia (green line).

**Table 1.** Measurements of the Tunica media, T. intima and T. adventitia of the sections before the carotid rete, in the carotid rete and after the rostral epidural rete mirabile and the thickness of the T. media as a percentage of the total thickness of the vessel.

|  | T. Media | T. Media + T. Intima | T. Media + T. Intima + T. Adventitia | T. Media Thickness: Total Thickness (%) |
|---|---|---|---|---|
| Pre-carotid rete | 0.53–0.60 mm | 0.55–0.63 mm | 0.63–0.73 mm | 83% |
| Carotid rete | 0.09–0.18 mm | 0.10–0.19 mm | 0.15–0.28 mm | 62% |
| Post-carotid rete | 0.28–0.33 mm | 0.30–0.35 mm | 0.43–0.68 mm | 54% |

From Table 1, it is evident that the Tunica media of the maxillary artery before it enters the rostral epidural rete mirabile and the tunica media of the intracranial maxillary artery at the exit of the rostral epidural rete mirabile are approximately three times thicker in comparison to the T. media of the rostral epidural rete mirabile arteries. We expected that the pre-carotid vessel's tunica media would be larger than the T. media of the rete for more regulation pre-rete.

*3.4. Pressure and Flow Experiment*

Flow rate:

The average flow rate of water through each different length of simulated rostral epidural rete mirabile is shown in Table 2. This table clearly illustrates that the flow rate decreased when the rostral epidural rete mirabile length increased. Table 3 confirms this by linear regression statistics through various lengths of the carotid rete network.

**Table 2.** In one minute, the average flow rate through the simulated rostral epidural rete mirabile network of varying lengths expresses as milliliters per minute.

| CR Length (m) | mL Per Minute (mL/min) |
|---|---|
| 6 | 275.00 |
| 5 | 400.00 |
| 4 | 400.40 |
| 3 | 428.00 |
| 2 | 510.40 |
| 1 | 575.00 |
| 0.5 | 647.80 |

**Table 3.** Linear regression statistics for flow rate through variable lengths of rostral epidural rete mirabile.

| Term | Estimate | Std. Error | *t*-Statistic | *p*-Value |
|------|----------|------------|---------------|-----------|
| Intercept | 643.347 | 9.682 | 66.450 | $p < 0.001$ |
| Length | −58.922 | 2.682 | −21.97 | $p < 0.001$ |

Linear regression was performed through the measurements. The fit of the measurements is illustrated in Figure 6.

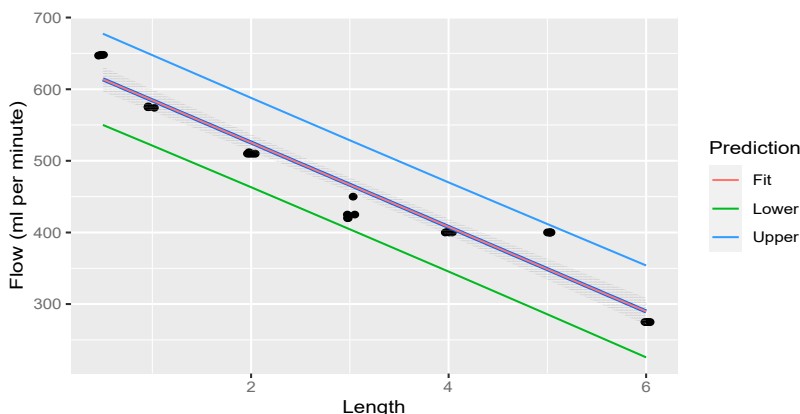

**Figure 6.** Linear regression through measurements illustrating flow rate decreasing with an increase in rostral epidural rete mirabile length. The shaded area indicates the 95% confidence region of the regression line, and the outer lines indicate the 95% prediction interval for observations.

The average pressure of water through each different length of rostral epidural rete mirabile is shown in Table 4. Pressure decreased as the rete length increased.

**Table 4.** Average pressure measured post the rostral epidural rete mirabile network of varying lengths in one minute, expressed as millimeters of mercury, at the head-up and head-down positions.

| Rete Length (m) | Average Pressure (mmHg) Head-Up | Average Pressure (mmHg) Head-Down |
|-----------------|--------------------------------|-----------------------------------|
| 6 | 313.53 | 157.51 |
| 5 | 298.52 | 150.01 |
| 4 | 286.52 | 142.51 |
| 3 | 277.52 | 136.51 |
| 2 | 264.02 | 133.51 |
| 1 | 256.52 | 127.51 |
| 0.5 | 238.52 | 120.01 |

Linear regression was performed through the measurements. The fit of the measurements is illustrated in Figure 7a,b. Table 5 illustrates linear regression statistic of pressure at variable lengths of rostral epidural rete mirabile, for the head-up position, with pressure starting at 330.027 mmHg. Table 6 illustrate linear regression statistic of pressure at variable lengths of rostral epidural rete mirabile, for the head-down position, with pressure starting at 180.015 mmHg.

From the data obtained in the experiment, we found that both the flow rate and pressure were indirectly proportional to rete length. The meshwork structure of numerous arteries of the rostral epidural rete mirabile reduces the flow rate of blood and blood pressure as it moves through the network to subsequently enter the brain.

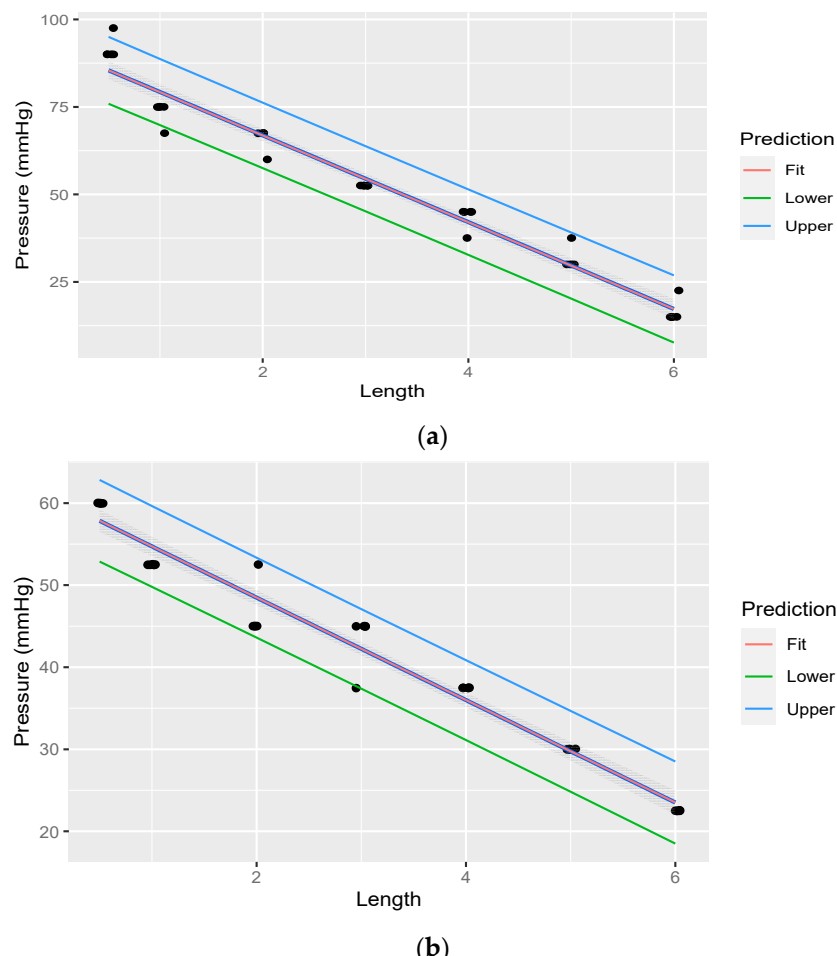

(**a**)

(**b**)

**Figure 7.** (**a**). Linear regression through measurements illustrating pressure decreasing with an increase in rostral epidural rete mirabile length, with head-up, with pressure measured post the rostral epidural rete structure. The shaded area indicates the 95% confidence region of the regression line, and the outer lines indicate the 95% prediction interval for observations. (**b**). Linear regression through measurements illustrating pressure decreasing with an increase in rostral epidural rete mirabile length, with head-down, with pressure measured post the rostral epidural rete structure. The shaded area indicates the 95% confidence region of the regression line, and the outer lines indicate the 95% prediction interval for observations.

**Table 5.** Linear regression statistic of pressure at variable lengths of rostral epidural rete mirabile, for the head-up position, with pressure starting at 330.027 mmHg.

| Term | Estimate | Std. Error | *t*-Statistic | *p*-Value |
|---|---|---|---|---|
| Intercept | 91.650 | 1.451 | 63.184 | $p < 0.001$ |
| Length | −12.396 | 0.402 | −30.856 | $p < 0.001$ |

**Table 6.** Linear regression statistic of pressure at variable lengths of rostral epidural rete mirabile, for the head-down position, with pressure starting at 180.015 mmHg.

| Term | Estimate | Std. Error | *t*-Statistic | *p*-Value |
|---|---|---|---|---|
| Intercept | 60.963 | 0.756 | 80.668 | $p < 0.001$ |
| Length | −6.243 | 0.209 | −29.825 | $p < 0.001$ |

The implication is that the structure of the rete only functions when blood flows. The pressure before, within, and after the rete is equal if there is no flow. Pressure decrease

occurs due to (a) a decrease in the diameter of the vessel it flows through and (b) resistance against the inside area of the vessel it flows through. The resistance depends on the (a) change in vessel wall diameter and (c) the change in momentum of laminar flow.

In the rete, the resistance does not change due to a decrease in vessel wall diameter, as shown in Table 1. An example of a pressure drop due to a decrease in vessel wall diameter is in the tibialis artery, where the pressure drops due to the reduction in flow. We calculated the area of a 10-mm vessel (Table 7) by the following equation:

$$A = \pi r^2$$

where:
    $A$ = Area
    $\pi$ = 3.14
    $r$ = radius

**Table 7.** Area of a 10 mm-vessel with a constant radius.

| Radius | Area (mm$^2$) | Number of Vessels | Total Area (mm$^2$) |
|--------|---------------|-------------------|---------------------|
| 5 | 78.54 | 1 × 10 | 78.54 |
| 2.5 | 19.64 | 2 × 5 | 39.28 |
| 2.5 | 19.64 | 4 × 5 | 78.56 |

We can confidently say that the rete arteries bifurcate in such a way that the total area of each vessel increases; the pressure drop and then the volume or flow does not change (Figure 8a,b).

The following equation was used to calculate the circumference of the artery (Table 8):

$$C = 2\pi r^2$$

where:
    $C$ = circumference
    $\pi$ = 3.14
    $r$ = radius

**Table 8.** The circumference of a 10-mm vessel with a constant radius.

| Radius | Circumference | Number of Vessels | Total Circumference |
|--------|---------------|-------------------|---------------------|
| 5 | 31.42 | ×1 | 31.42 |
| 2.5 | 15.70 | ×4 | 62.80 |

These data suggest that, consequently, the resistance due to the friction of the vessel wall is double in the bifurcation. An area offers resistance in the vessel through which blood flows, doubles and thus gives the friction coefficient, resulting in a pressure drop.

This, in turn, would mean that the resistance in the rete is influenced by the viscosity of the blood or fluid and the resistance exerted by the vessel wall.

The rostral epidural rete mirabile structure, with its numerous bends and turns, influences pressure by the law of laminar flow. The following factors all affect the flow and thus the pressure in a vessel:

$$\Delta P = \frac{1}{2} f_s p u^2 \frac{\pi R b}{D}\, \theta / 180° + \frac{1}{2} K b p u^2$$

where:
    $f_s$ = Moody friction factor
    $p$ = density
    $u$ = mean flow velocity

$Rb$ = bend radius
$D$ = tube diameter
$\Theta$ = bend angle
$Kb$ = bend loss coefficient

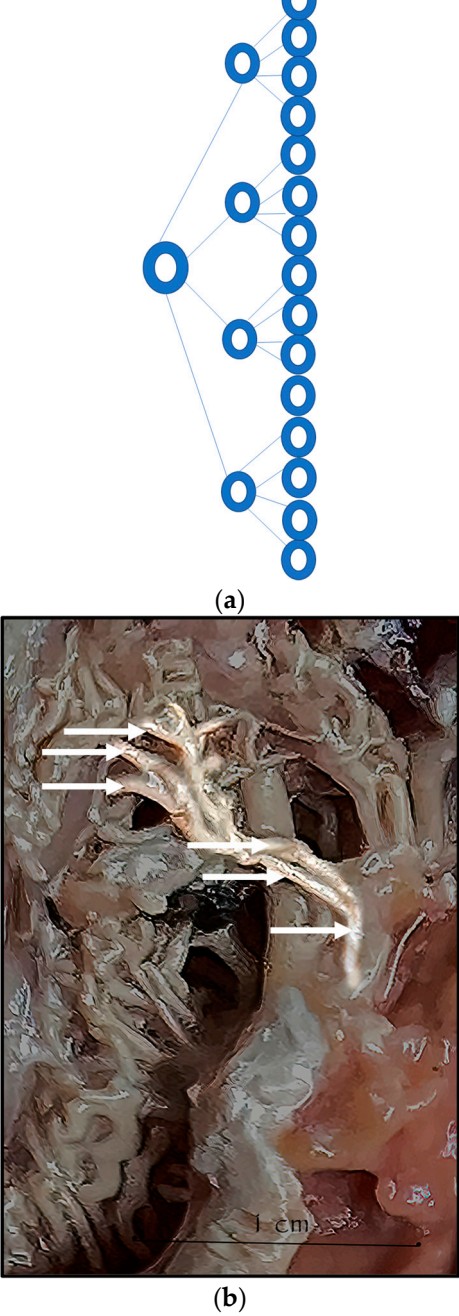

(**a**)

(**b**)

**Figure 8.** (**a**). Schematic presentation of bifurcation of the rostral epidural rete mirabile arteries. (**b**). Highlighted arteries of the rostral epidural rete mirabile indicating bifurcation (marked by white arrows) with arteries splitting into several arteries remaining similar in size.

Centripetal force is the force a rotating object experiences. The path of the centripetal force is in the direction of the pivot point or inward-directed to the middle of the circle. The giraffe moves its head from ground level to fully direct it in a circular path (Figure 9), similar to an object moving in a circle from the bottom to the top of the circle. The head experiences a centripetal force directed caudally along the atlanto-occipital joint. Blood that

moves towards the head as it is moved to ground level experiences a centripetal force away from the head. The blood is forced to remain towards the head area, instead of flowing back towards that heart as expected due to gravity.

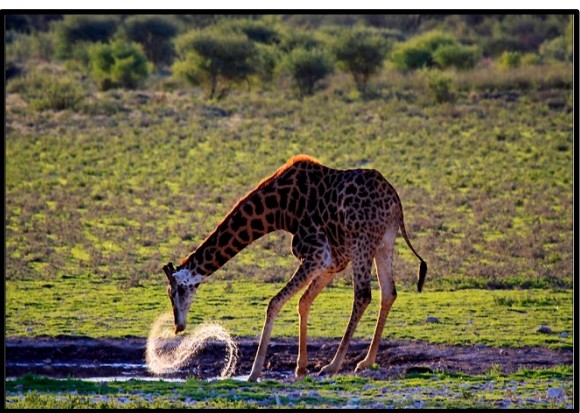

**Figure 9.** Illustration of movement of giraffe head from ground level to erect in a circular path towards the head area, instead of flowing back towards that heart as expected due to gravity. Centripetal force is calculated as per the below formula (Photo: F. Deacon).

The speed at which the giraffe moves its head from ground level to fully erect is currently unknown and, unfortunately, not documented anywhere. Therefore, we can only speculate on the additional effect of the centripetal force on the cerebral blood supply in the giraffe.

## 4. Discussion

Our study on the blood supply from the heart to the brain in the physiologically challenged giraffe (*G. camelopardalis*) reveals the unique functional significance of the rostral epidural rete mirabile.

Ref. [31] showed that the rostral epidural rete mirabile arteries in dromedary camels (*C. dromedarius*) measured 305 ± 9.7 cm, indicating the importance of the length and increased surface area for influencing the regulation of blood flow and, subsequently, blood pressure. Unfortunately, the latex mixture in our study did not fill the rete structure. Due to its extremely fine and interconnected network structure, we could not dissect the rostral epidural rete mirabile and measure its length. However, observing the rete structure closely revealed the network of numerous bends, suggesting a definite influence on blood flow through resistance exerted by bends in arterial walls [32,33]. The interconnected meshwork of arteries adds an overall significant enlargement of the cross-sectional area in vessels, resulting in reduced resistance as conferred using Poiseuille's Law [9,32]. The structure will subsequently also reduce pressure [34]. In a study on the cerebral arterial blood flow in the Yak (*Bos grunniens*), the rostral epidural rete mirabile was found to be highly developed compared to the rostral epidural rete mirabile in Chinese cattle (*Bos taurus*). In the Yak, the rete enables more efficient blood exchange and fulfills the critical function of a blood reservoir for blood supply to the brain [14]. Yak are well known to reside at high altitudes with characteristically low oxygen availability. This species' rostral epidural rete mirabile is essential to efficiently shield, transport and stabilize blood flow and oxygen to the brain [14].

Similarly, the alpaca (*Vicugna pacos*) resides at high altitudes with low oxygen availability and endure extreme ambient temperature variability. It is speculated that the rostral epidural rete mirabile in the alpaca might aid in coping with hypoxia and extreme environmental temperature fluctuations [9]. This brings us back to our initial research question: Does the rostral epidural rete mirabile similarly fulfill this function in the giraffe, with its extreme head-to-heart distance aiding as a reservoir for brain perfusion and buffering blood

rushing to the brain at the different head positions? This deduction was verified by our experiment, showing that rostral epidural rete mirabile length is indirectly proportional to both blood flow and pressure. The rostral epidural rete mirabile is, therefore, automatically altering the hemodynamics of cerebral blood flow using its structure.

Studies conducted on selective brain cooling in artiodactyls by [17,21,35] describe the inimitable position of the rostral epidural rete mirabile where it lies in the cavernous sinus, at the base of the brain; indeed, it is positioned below the brain, within the cranium [4,36]. The position of the rostral epidural rete mirabile in the brain is significant regarding brain cooling and supplying blood efficiently to the brain [14,17,36]. Correspondingly, this position in the giraffe further supports the idea that the rostral epidural rete mirabile influences blood flow and pressure to and from the brain, which is especially important with the quick movement and position change of the head to ground level and again to a fully erect position. In a giraffe, the alternative pathway of blood to the brain via the vertebral arteries and the basilar artery that enters the circle of Willis in other animals does not exist. The basilar artery in the giraffe does not assist in the blood supply to the brain [8,19,21]. The maxillary artery through the rostral epidural rete mirabile is the dominant vessel supplying blood to the brain in giraffes.

Another observation in a giraffe that differs from other Artiodactyls and Perissodactyls is the heart's position concerning the brain. The pathway from the heart to the brain is notably straight, lowering resistance to flow and relieving pressure from the heart pumping 2 m against gravity to the brain. In a study on giraffes, [5] measured carotid blood flow and calculated the resistance to flow at head-up and head-down positions. Their results showed that when the head is raised, extracranial vasoconstriction takes place to prevent fainting. When the head is down, they suggest that blood flow to the brain is unrestricted. When looking at the position of the heart and the pathway from the heart to the brain, a notable straight path can be observed in comparison to other animals in which the heart is situated at an angle to the brain. The angle creates a bent path for blood flow from the heart to the brain, increasing arterial resistance [32,37,38]. The 2-m head-to-heart pathway of blood in giraffes has less resistance than, for example, sables and elephants' pathways due to the straight pathway. This supports the idea of little or no control of blood flow to the brain in the head-down position. The observed notable bending of the maxillary artery before entering the rostral epidural rete mirabile in an otherwise straight pathway creates additional resistance to blood flow rushing to the brain when the head is lowered. The twisted arteries of the rostral epidural rete mirabile, the significant length of the rete arteries and the notable bend at the maxillary before entrance into the rete all in conjunction provide resistance to flow and therefore aid in the prevention of brain damage with the quick bending down from a fully erect to a ground position.

Histologically, the maxillary artery prior to entering the rostral epidural rete mirabile structure and the intracranial maxillary artery post the rete are characteristically thicker-walled than the rete arteries. The muscular content of the rostral epidural rete mirabile arteries enables a greater extent of control. Again, resistance against blood flow is exerted by structural features, with blood moving from a thicker maxillary to the thinner rete arteries and again through the thicker intracranial maxillary into the circle of Willis. The rostral epidural rete mirabile arteries contain dense, smooth muscle indicating its ability to contract and expand. Expansion of the rostral epidural rete mirabile arteries will be restricted due to the limited space where it lies within the cavernous sinus, enclosed by bone. However, contraction of the rostral epidural rete mirabile network arteries will be possible due to the high muscular content. The ability of a large thick artery to contract in a limited space is restricted; however, the contraction ability of a network of various small arteries is more remarkable [39].

We suggest that the pool of blood present within the rostral epidural rete mirabile meshwork, situated just beneath the brain, helps prevent the giraffe from fainting together with the carotid-vertebral anastomotic connection when the head is moved in a quick motion from ground level to fully erect. It is essential that the brain receives a continuous

flow of blood and thus oxygen [10,24,33]. The blood in the rete can be pushed via the contraction of the rostral epidural rete mirabile arteries into the circle of Willis. Additionally, when the head is moved from ground level to fully erect, the force of gravitation pulls blood away from the brain, and the carotid artery enlarges with a parallel drop in pressure. The vertebral arteries are neighbored by connective tissue and back muscles as it courses through the transverse foramina and through the axis and atlas, avoiding enlargement. At this point, the vertebral arteries connect to the anastomotic artery that subsequently connects to the carotid artery. During the loss of pressure in the carotid artery, blood can be moved from the vertebral arteries via the anastomotic artery into the carotid artery that transitions into the maxillary artery, pushing blood into the rostral epidural rete mirabile and the circle of Willis. This pathway requires less pressure to be generated by the heart to supply adequate blood quickly to the brain during the movement of the head to a fully erect position. Fainting due to a loss of continuous blood flow to the brain is negated by the immediate source of blood to the cerebral arterial circle originating from the vertebral artery via the anastomotic artery into the carotid artery, as well as a small volume of blood contained within the rete arterial structure. The retained blood volume in the rete and the vertebral blood supply contributes to and ensures the crucial continuous flow of blood and thus oxygen to the brain to prevent fainting or even an ischemic cascade, owing to too little blood present in the brain [40]. This pathway of cerebral blood supply aid in temporary relief during the split seconds time-lapse, allowing the nervous system to instigate the heart to pump blood against gravity to reach the brain. Even though nervous stimulation is instant, a time-lapse of a few seconds occurs before the heart can pump blood to the brain in a fully erect position. A study by [40] on blood flow circulation in intracranial networks showed that due to the elasticity and subsequent cross-sectional area of arterial blood vessels, a relative blood volume is partly retained, slowing the pulsatile flow. The retained blood is released using blood vessel contraction [40]. The centripetal force exerted on the blood accumulated towards the head also adds to blood availability to the brain during the movement of the head from ground level to fully erect, at a critical moment, i.e., milliseconds.

Additionally, [35] investigated selective brain cooling and the influence of the rostral epidural rete mirabile in sheep. They concluded that brain blood flow and brain metabolic heat production differed contrary to the expected close relationship, as previously verified by [41]. Can this variation to the desired close relationship be due to the effect exerted by the rostral epidural rete mirabile structure on blood flow?

The combination of the results in our study gives a clear indication of the functional significance and efficiency of the rostral epidural rete mirabile in giraffes, as follows: (a) the structural arterial meshwork with characteristic turns and bends, and subsequently increased artery length through which blood flow must proceed before entrance into the brain, exert resistance to blood racing to the brain when the head is dropped to the ground; (b) the rostral epidural rete mirabile structure is positioned just beneath the brain, retaining a minimal amount of blood; (c) the brain is supplied mainly by the maxillary artery through the rostral epidural rete mirabile, with a rudimentary basilar artery not contributing to the brain blood supply and (d) the resistance to blood flow due to the structure and position of the rete when the head is in the upright position is counteracted by the unique carotid-vertebral anastomosis allowing an immediate cerebral blood supply.

## 5. Conclusions

The structure, function and position of the rostral epidural rete enable adequate circulation to the brain in the long-necked giraffe. Future research can focus on determining its functionality in other long-necked artiodactyls such as the llama (*Lama glama*), vicuna (*Vicugna vicugna*) and alpaca (*V. pacos*). Additionally, do fossil giraffids, although with characteristically shorter necks, also have a rostral epidural rete mirabile to assist with physiological challenges associated with a long head-to-heart distance? The rostral epidural rete mirabile in the giraffe is one of a collection of features that, in combination, enable

efficient brain perfusion and prevention of brain aneurisms despite its peculiar build resulting in numerous physiological challenges.

**Author Contributions:** Conceptualisation: M.S.v.d.W., W.D. and F.D.; methodology: M.S.v.d.W.; formal analysis: S.v.d.M.; investigation: M.S.v.d.W.; writing—original draft: M.S.v.d.W.; Writing—Review and editing: F.D., W.D., J.G. and S.v.d.M. All authors have read and agreed to the published version of the manuscript.

**Funding:** This project was funded by the Kroonstad Dierehospitaal and Midlands Veterinary Wholesalers Pty Ltd. This work is based on the research supported wholly/partially by the National Research Foundation (NRF) of South Africa (Grant Number: RA201126576714).

**Institutional Review Board Statement:** Ethics approval nr. UFS-AED2020/0083. Approval was obtained from the Animal Ethics Research Committee at the UFS, SPCA and DESTEA.

**Informed Consent Statement:** Not applicable.

**Data Availability Statement:** All data will be available from the main author.

**Acknowledgments:** Kroonstad Dierehospitaal for the use of the animal hospital premises, and their staff, especially Ben Koloti and Daniel Mthembu, for assistance with experimental work. Midlands Veterinary Wholesaler Pty Ltd. for assistance in the funding of the project. This work is supported on the research wholly / in part by the National Research Foundation (NRF) of South Africa (Grant Number: RA201126576714).

**Conflicts of Interest:** The authors declare no conflict of interest.

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
