# Peer review of "The Rostral Epidural Rete Mirabile: Functional Significance in Blood Flow Regulatory Mechanisms in Giraffe (Giraffa camelopardalis)"

_2813-0545, doi:10.3390/anatomia2020013_

Round 1

Reviewer 1 Report

This is a very interesting article tackling the vascular anatomy of the giraffe.

I congratulate you on this experimental analysis of the blood flow.

However, there are some aspects that require improvement:

1.      In the legend of the microscopic figures you need to include the type of staining and the magnification level.

2.      Figure 9 please credit it to the original photographer.

3.      Please insert a Conclusion section in which to underline the possible use of this research in future anatomical models and medical devices.

4.      At the end of the manuscript, you should insert an author contribution section according to MDPI formatting.

5.      Also please format the references according to MDPI standards.

Author Response

RESPONSE TO REVIEWERS COMMENTS

Open Review

Quality of English Language

( ) English very difficult to understand/incomprehensible
( ) Extensive editing of English language and style required
( ) Moderate English changes required
(x) English language and style are fine/minor spell check required
( ) I am not qualified to assess the quality of English in this paper

Yes

Can be improved

Must be improved

Not applicable

Does the introduction provide sufficient background and include all relevant references?

(x)

( )

( )

( )

Are all the cited references relevant to the research?

(x)

( )

( )

( )

Is the research design appropriate?

(x)

( )

( )

( )

Are the methods adequately described?

(x)

( )

( )

( )

Are the results clearly presented?

(x)

( )

( )

( )

Are the conclusions supported by the results?

( )

(x)

( )

( )

Comments and Suggestions for Authors

This is a very interesting article tackling the vascular anatomy of the giraffe.

I congratulate you on this experimental analysis of the blood flow.

However, there are some aspects that require improvement:

  1. In the legend of the microscopic figures you need to include the type of staining and the magnification level.

*Response to comment 1: Added staining type and magnification level

  1. Figure 9 please credit it to the original photographer.

*Response to comment 2: Added F. Deacon, the original photographer to the legend of the picture

  1. Please insert a Conclusion section in which to underline the possible use of this research in future anatomical models and medical devices.

*Response to comment 3: Added conclusion in lines 577-582

  1. At the end of the manuscript, you should insert an author contribution section according to MDPI formatting.

*Response to comment 4: Added author contributions lines 587-590

  1. Also please format the references according to MDPI standards.

*Response to comment 5: Altered references in manuscript and in reference list according to MDPI Standards

Submission Date

28 January 2023

R2

 Top of Form

Open Review

Quality of English Language

( ) English very difficult to understand/incomprehensible
( ) Extensive editing of English language and style required
( ) Moderate English changes required
(x) English language and style are fine/minor spell check required
( ) I am not qualified to assess the quality of English in this paper

Yes

Can be improved

Must be improved

Not applicable

Does the introduction provide sufficient background and include all relevant references?

(x)

( )

( )

( )

Are all the cited references relevant to the research?

(x)

( )

( )

( )

Is the research design appropriate?

(x)

( )

( )

( )

Are the methods adequately described?

(x)

( )

( )

( )

Are the results clearly presented?

( )

( )

( )

( )

Are the conclusions supported by the results?

(x)

( )

( )

( )

Comments and Suggestions for Authors

interesting topic which is well-researched.  could the vasculature be explored using a CT/MRI rather than from dead animals. I understand that there was the ethical clearance however this type of research could be objectionable by certain groups 

Submission Date

28 January 2023

Date of this review

06 Feb 2023 18:21:55

Bottom of Form

© 1996-2023 MDPI (Basel, Switzerland) unless otherwise stated

*Response to comment Reviewer 2: MRI/CT scanning can be an option for smaller animals. For animals such as Giraffe, the only option in South Africa is the use of hunted animals.

R3

Open Review

Quality of English Language

( ) English very difficult to understand/incomprehensible
( ) Extensive editing of English language and style required
( ) Moderate English changes required
(x) English language and style are fine/minor spell check required
( ) I am not qualified to assess the quality of English in this paper

Yes

Can be improved

Must be improved

Not applicable

Does the introduction provide sufficient background and include all relevant references?

(x)

( )

( )

( )

Are all the cited references relevant to the research?

(x)

( )

( )

( )

Is the research design appropriate?

(x)

( )

( )

( )

Are the methods adequately described?

(x)

( )

( )

( )

Are the results clearly presented?

(x)

( )

( )

( )

Are the conclusions supported by the results?

(x)

( )

( )

( )

Comments and Suggestions for Authors

The manuscript (ID: anatomia-2207979) titled “The rostral epidural rete mirabile: Functional significance in blood flow regulatory mechanisms in giraffe (Giraffa camelopardalis)”  is an interesting paper that provides new information regarding the anatomical characterization and physiological functions of the rostral epidural rete mirabile (rete mirabile epidurale rostrale), in the giraffe (Giraffa camelopardalis).

The entire manuscript is well organized and structured.

The experimental design and analyses are comprehensively described and applied. Furthermore, the results are clearly presented, very interesting and fairly discussed in the context of your findings.

The methods and techniques used are sound and appropriate for the study and the results are very interesting.

I kindly suggest some additional efforts (minor revision) to improve your manuscript.

Finally, I believe that suggested changes will contribute to improve the entire manuscript, making it appropriate for publication in the journal "Anatomia".

 The manuscript should be revised by addressing the following points:

Abstract section:

The Abstract should include the background the context, its aims and the purpose of the study. Moreover, it must include the main results and the conclusions as a brief summary and potential implications. Please write the abstract following these suggestions.

*Response to comment on Abstract section: Abstract was altered according to the above suggestions

Results section:

Line 235-239. The authors observed that “Having access to the complete giraffe, we observed that the heart is situated anteriorly in the chest compared to the heart of other ruminants, such as elephants and sable antelope (Figure 4).”

Why did you choose to compare giraffes, elephants and antelopes? Is the position of the heart in ruminants such as Bos taurus different?

Have you observed the position of the heart in cetacean species?

Have you observed the position of the heart in the llama or alpaca?

*Response to comment on Line 235-239: Elephants were chosen because of it being a mammal with large body size, with a higher blood pressure in order to efficiently circulate blood

Sable antelope were used, as an example of an antelope that has a medium-length neck and is of smaller body size than an Elephant or Giraffe. The antelope will have a normal blood pressure, without physiological challenges faced by elephant, due to large body size and specifically by giraffe, due to the long head-to-heart distance.

Bos taurus has a similar pathway as the sable antelope.

Alpaca has a similar pathway as the giraffe.

The pathway in the whale, appears relatively straight as in giraffe

Figure 3a, 3b: Pleas add the scale bar or the magnification of the images.

*Response to comment on Figure 3a,3b: The scalebar was added into both the images

Figure 5a, 5b and 5c can be grouped in one figure named figure 5 and the letters a, b, c indicating the corresponding figure, should be added in the correspondent images. Pleas add the scale bar or the magnification of the images.

*Response to comment on Figure 5a, b, c: Grouped the three images together and added the magnification

Figure 8b. Please increase the image quality.

*Response to comment on Figure 8b: Increased image quality

Line 360: In the figure legend you wrote “…indicating bifurcation….”. Please indicate the bifurcations in the figure by arrows.

*Response to comment on Line 360: Added white arrows to indicate bifurcation

Submission Date

28 January 2023

Date of this review

05 Feb 2023 09:42:06

Reviewer 2 Report

interesting topic which is well-researched.  could the vasculature be explored using a CT/MRI rather than from dead animals. I understand that there was the ethical clearance however this type of research could be objectionable by certain groups 

Author Response

(The authors gave the same response as above.)

Reviewer 3 Report

The manuscript (ID: anatomia-2207979) titled “The rostral epidural rete mirabile: Functional significance in blood flow regulatory mechanisms in giraffe (Giraffa camelopardalis) is an interesting paper that provides new information regarding the anatomical characterization and physiological functions of the rostral epidural rete mirabile (rete mirabile epidurale rostrale), in the giraffe (Giraffa camelopardalis).

The entire manuscript is well organized and structured.

The experimental design and analyses are comprehensively described and applied. Furthermore, the results are clearly presented, very interesting and fairly discussed in the context of your findings.

The methods and techniques used are sound and appropriate for the study and the results are very interesting.

I kindly suggest some additional efforts (minor revision) to improve your manuscript.

Finally, I believe that suggested changes will contribute to improve the entire manuscript, making it appropriate for publication in the journal "Anatomia".

 The manuscript should be revised by addressing the following points:

Abstract section:

The Abstract should include the background the context, its aims and the purpose of the study. Moreover, it must include the main results and the conclusions as a brief summary and potential implications. Please write the abstract following these suggestions.

Results section:

Line 235-239. The authors observed that “Having access to the complete giraffe, we observed that the heart is situated anteriorly in the chest compared to the heart of other ruminants, such as elephants and sable antelope (Figure 4).”

Why did you choose to compare giraffes, elephants and antelopes? Is the position of the heart in ruminants such as Bos taurus different?

Have you observed the position of the heart in cetacean species?

Have you observed the position of the heart in the llama or alpaca?

 Figure 3a, 3b: Pleas add the scale bar or the magnification of the images.

 Figure 5a, 5b and 5c can be grouped in one figure named figure 5 and the letters a, b, c indicating the corresponding figure, should be added in the correspondent images. Pleas add the scale bar or the magnification of the images.

Figure 8b. Please increase the image quality.

Line 360: In the figure legend you wrote “…indicating bifurcation….”. Please indicate the bifurcations in the figure by arrows.

Author Response

(The authors gave the same response as above.)
